# Breast Reconstruction after Mastectomy with the Use of an Implant and Serratus Anterior Fascia Flap—Initial Clinical Evaluation

**DOI:** 10.3390/jpm11111142

**Published:** 2021-11-03

**Authors:** Mauro Tarallo, Federico Lo Torto, Fabio Ricci, Paolo Dicorato, Francesco Luca Rocco Mori, Federica Vinci, Paola Parisi, Manfredi Greco, Carlo De Masi, Alessandra Rita La Manna, Silvia Piroli, Diego Ribuffo

**Affiliations:** 1Department of Surgery “P. Valdoni”, Sapienza University of Rome, 00161 Rome, Italy; mauro.tarallo@uniroma1.it (M.T.); paolo.dicorato@outlook.it (P.D.); francescolucarocco.mori@uniroma1.it (F.L.R.M.); federica.vinci@uniroma1.it (F.V.); paola.parisi@uniroma1.it (P.P.); diego.ribuffo@uniroma1.it (D.R.); 2Breast Unit, Santa Maria Goretti Hospital, Sapienza University, 04100 Latina, Italy; fa.ricci@ausl.latina.it (F.R.); c.demasi@ausl.latina.it (C.D.M.); al.lamanna@ausl.latina.it (A.R.L.M.); s.piroli@ausl.latina.it (S.P.); 3Department of Plastic Surgery, University of Catanzaro Hospital, 88100 Catanzaro, Italy; manfredigreco@unicz.it

**Keywords:** breast cancer, plastic surgery, breast reconstruction, serratus fascia flap

## Abstract

Prosthesis-based techniques are the predominant form of breast reconstruction worldwide. The most performed surgical technique involves the placement of the expander in a partial submuscular plane. The coverage of the implant remains a difficult management problem that can lead to complications and poor outcomes. The use of the serratus fascia flap may be the best choice to create a subpectoral pocket for the placement of a tissue expander, with excellent results in terms of morbidity and cost-effectiveness. A total of 20 breast reconstructions with the inferolateral coverage with the serratus fascia were performed. Patients demonstrated a low overall complication rate (9.5%), such as seroma and infection, with complete resolution during the follow-up and no major complications. The US examination of the soft tissues over the implant reported thickness measurements that demonstrated a good coverage over the inferolateral area. Our study shows that using the serratus fascia flap to create a pocket with the pectoralis major for the placement of the tissue expander is an effective technique during two-stage breast reconstruction. The resulting low rate of morbidity and the US findings collected reveal the safety of this procedure. Its success relies on appropriate patient selection and specific intraoperative technique principles.

## 1. Introduction

Breast reconstruction is nowadays an integral part of breast cancer treatment. Immediate reconstruction in affected women is accompanied by a lower incidence of postoperative psychological morbidity related to loss of the breast [1]. Conservative approaches in mastectomy allow immediate reconstruction with good aesthetic outcomes.

Immediate reconstruction with autologous tissue is preferred in selected cases while breast implant reconstruction is more widely indicated [2]. Compared with autologous reconstruction, implant reconstruction offers a quicker surgery, reduces donor site morbidity, and shortens post-operative recovery time. Immediate one-stage direct to implant (DTI) breast reconstruction has been recently improved by the introduction of biological and synthetic matrices [3,4]. The two-stage option of expander breast reconstruction followed by permanent implant positioning remain commonly performed following mastectomy [5]. Nowadays, the most common surgical technique for expander positioning is the partial submuscular one, as described by Serra-Renom et al. [6], after abandoning the complete submuscular placement, which leads to an unnatural appearance of the breast due to poor lower breast pole expansion with complete muscular coverage. Partial submuscular expander positioning leaves the inferolateral aspect of the implant coverage to solve. Several techniques have been used to achieve complete coverage and control of the muscular pocket [7,8,9,10,11], including securing the pectoralis muscle to the lower mastectomy skin flap, using de-epithelialized dermal flaps, elevating the adjacent serratus muscle (SM), using pectoralis minor flaps, recruiting rectus and external oblique fascia, and, more recently, using acellular dermal matrices [12,13]. Soft tissue-based methods can be effective, but they may increase postoperative pain or compromise functional outcomes. ADM (acellular dermal matrix)-assisted breast reconstruction remains controversial within the literature due to an increased rate of postoperative complications in several large case series. These include elevated risks of infection, seroma, and reconstructive failure [14,15].

One important technique that is seemingly underused and not well studied involves the elevation of the serratus anterior fascia to control the inferolateral border of a breast prosthesis. The use of the serratus fascia (SF) provides the advantages associated with using natural, vascularized tissue, but potentially without the pain and functional compromise associated with complete muscle dissection [10].

This study evaluates clinical follow up and US findings in a group of patients who underwent mastectomy and immediate expander positioning using SF flap, assessing postoperative outcomes and the ability of serratus fascia to cover and preserve the inferolateral aspect of the implant rating soft tissue thickness over the implant.

## 2. Materials and Methods

We performed a prospective study in 18 patients treated for breast cancer in the certified breast unit of Latina Hospital between November 2018 and October 2019 with two stage reconstruction. The study was approved by our institutional review board and appropriate informed consent was obtained from all the patients. Skin sparing (SS) mastectomy by a superolateral incision was carried out by a single senior breast surgeon and all patients benefited from immediate tissue expander reconstruction by the plastic surgeon. A tissue expander was placed in a musculofascial pocket which included the pectoralis major, the serratus anterior fascia, and the superficial pectoral fascia inferiorly to completely cover the implant (Figure 1). Closure of the pocket was done by approximating the fascial flap to the lateral and inferior border of the pectoralis major muscle with 3/0 vicryl sutures. The total number of breast mounds reconstructed was 20. Tissue expansion started intraoperatively with 50 mL of saline solution for all reconstructed breasts, then subsequent injections were done in the clinic every 1–3 weeks. Drains were kept until the daily collection became less than 30 mL/day. All these patients were evaluated after at least one expansion with an ultrasound (US) examination and they were subjected to a VAS score pain evaluation the day after and the third day after surgery (Figure 2). Patients’ demographic data were collected, which included patient age, BMI, comorbidity, smoking, breast side, the time of postoperative expansion start, duration of postoperative drain use, carried out neoadjuvant chemotherapy and radiotherapy, and development of complications, which include seroma, hematoma, skin necrosis, wound dehiscence, infection, expander loss, capsular contraction, malposition, and any other complications (Table 1). At the time of the US evaluation, sonographic measurements were collected, i.e., skin and soft tissue thickness, echogenicity, and thickness of serratus anterior fascia in the most lateral and inferior aspect of the reconstructed breast mound. Since the pectoralis major muscle appears hypoechoic and the fascia hyperechoic, the thickness was measured inferiorly, and the oblique line was measured superiorly and laterally oriented, corresponding to a sharp shift in echogenicity and thickness over the implant, which probably matches the approximation line between the fascial flap and the inferolateral border of the pectoralis major muscle. In addition, we considered the sonographic evidence of different local complications, such as the presence of periprosthetic fluid, inhomogeneities of soft tissues, liponecrosis, hematoma, seroma, infection, and lymphoceles.

## 3. Results

We performed 20 expander-based breast reconstructions after mastectomy in 18 patients (2 patients had bilateral reconstruction). In total, 12 patients performed neoadjuvant chemotherapy, 2 patients underwent postoperative RT, and 1 underwent postoperative CHT (cyclophosphamide, methotrexate 5-FU). A total of 6 of them had comorbidities, including obesity, diabetes, and hypertension, and 7 of them were smokers. The mean follow-up period was 17.45 months (12–23). Patient ages ranged between 37–62 (mean 48) and the average BMI was 26 (21–35). The average time of postoperative permanence drain was 15 days. The overall complication rate was 9.5%, including seroma and hematoma.

We did not have any major complication or reconstruction failure and any case of capsular contracture was observed during the follow up-period. The mean time of US examination was 6.8 months (3–15 months).

Thickness measurements over the implant was divided as follows: mean skin and subcutaneous thickness was 9.21 mm, mean serratus fascia thickness was 6.37 mm, and mean skin and soft tissue total implant coverage was 15.59 mm (Table 2).

In most patients, the tissue expander was folded and had an irregular profile; in 10 breasts, the presence of periprosthetic fluid was observed and the widest fluid collection was 30 × 17 mm. Any nodule of liponecrosis or of inflammatory nature was observed during US examination.

Regarding pain management after surgery, the systemic postoperative analgesic regimen was acetaminophen (1 g × 3/day); the mean VAS for the first day after surgery was 4 (moderate), and the mean VAS for the third day after surgery was 2 (mild). (Table 3) (Figure 3).

## 4. Discussion

Partial subpectoral expander coverage is, by definition, inadequate to cover the lower lateral part of the expander. Several techniques have been described to solve this challenge. Securing the lower edge of the pectoralis major muscle to the lower mastectomy skin flap provides muscle coverage in the upper part and leaves the lower pole free to expand in a subcutaneous plane [16]. Expander displacement can occur and stretching transferred by the expander directly on a thinned mastectomy skin flap can occasionally cause skin necrosis and eventual extrusion. These advantages make this option suitable in selected cases when adequate subcutaneous fat is preserved under the lower skin flap. Elevation of adjacent muscular flaps increases aesthetic outcomes, but is more invasive and improves postoperative pain. Firstly described by Saint-cyr et al. [17], the use of the serratus fascia for inferolateral coverage of tissue expander became more popular among breast reconstructive surgeon techniques in order to get comparable benefits and without higher complication rates of harvesting serratus muscle flap. The advantages of using serratus fascial flap are implant position control avoiding lateralization and definition of the lateral inframammary fold, allowing adequate inferior pole fullness or expansion, and at the same time offloading mechanical stress on the inferior skin envelope, particularly in case of vascular compromise of mastectomy skin flaps. It is an autologous well-vascularized barrier between the implant and the skin, shows better tolerance to postoperative radiation, and in comparison with serratus anterior muscle, results in less post-operative pain, drainage, and loss of mobility [17,18,19,20,21,22,23,24]. Although SF elevation shows similar complication rates to SM during the first stage breast reconstruction, Seth et al. [19] reported an absolute increase in postoperative hematomas and seromas in SM elevation, related to a more elaborate dissection of harvesting SM flap. Additionally, they showed greater intraoperative TE fill volumes in SF patients, resulting in fewer postoperative expansions needed to reach the final TE volume. Some limitations make the SF flap unreliable, as iatrogenic injury, anatomical attenuation in patients with low BMI, and patient comorbidities. In our study, the mean BMI was 26, in accordance with Saint-cyr; however, following the study by Bordoni et al. [20], we extended the indication to all patients undergoing two-stage reconstruction with a fascial plane preserved after mastectomy. We report an overall complication rate of 9.5%, comparable to previous studies. In order to evaluate the adequacy of the serratus fascia flap to cover the inferolateral aspect of the expander, standardized measurements of the skin-subcutaneous tissue and of the serratus fascia thickness were performed after at least one expansion. This means that at the time of US evaluation, a capsule had already been formed around the implant. As the serratus fascia is composed of dense connective tissue, such as the fibrous capsule, it is impossible to differentiate them, as they appear as a conjoined echoic line above the implant surface, rather than a stratified line. As reported by Gossner et al. [21], the most specific and objective US finding of capsule contracture is the measurable thickening of the fibrous capsule. Several studies described the thickness of a normal fibrous capsule around 1 mm and considered a thickness of more than 1.5 mm to be pathologic [25,26]. In the present study, we did not observe any case of capsular contracture. We believe that at the time of US evaluation, the rate of fibrous capsule on the conjoined fascia-capsula thickness was similar to a normal fibrous capsule. In addition, it is well known that disinsertion of sterno-costal fibers of PMM in breast surgery tipically bring to muscular atrophy of the lower parts covering the implant [27,28,29,30,31,32,33,34,35,36,37]. Knabben et al. [38] reported a significative decrease in the thickness of the PMM using standardized US periodical measurements. It is reasonable to conclude that the same process occurs detaching costal insertions of serratus muscle, making questionable the advantage of obtaining a major thickness coverage of the implant in the long term using SM. In our series, the US measurements of serratus fascia flap show good quality and thickness of tissue expander coverage in its infero-lateral part. The use of acellular dermal matrix (ADM) keeps stirring great interest and had become an integral part in implant-based breast reconstruction, but there seems to be no consensus regarding whether ADM use increases infection risk. Moreover, combining ADM with expander two-stage reconstruction is considered to be significantly more expensive than using autologous/tissue expander combinations [39]. Tissue integration of an ADM is related to vascular ingrowth of the matrix, and the CEUS in vivo evaluation by Parvizi et al. reported enhancement in the ADM after a month [40]. Ballesio et al. attested with a US examination partial integration of ADM at six months of follow-up and complete integration at 12 months. They also focused on ultrasonography changes of ADM in IBBR and US characteristics to distinguish benign lesions from tumor recurrence [41,42]. In effect, the intrinsic nature of ADM leads to an inflammatory response, which can lead to the development of inflammatory nodules within the breast and appear as a new palpable mass in patients with a history of breast cancer, sometimes requiring further diagnostic studies or invasive procedures to rule out recurrent disease [43,44]. This study had certain limitations. The lack of US periodical comparisons to establish changes of SF thickness as well as the lack of a control group can be major drawbacks. A randomized control trial, a large number of patients, and a longer follow-up should provide more definitive, high-level evidence to influence patient decision making, though results are complicated by anatomical and surgical variations of the SF after mastectomy.

## 5. Conclusions

The use of serratus fascial flaps combines the advantages of using muscular flaps with the advantages of using acellular allograft material by providing vascularized autologous tissue without violating adjacent muscles. As serratus fascia flaps are autologous, they do not lead to an inflammatory response, and inflammatory nodules can’t be used for differential diagnosis with suspicious palpable mass of recurrent disease. When present, elevation of SF for inferolateral coverage of TE is a safe, viable, autologous, and cost-effective option in expander-based breast reconstruction. Achieving good outcomes in terms of aesthetics, post-operative complications, and women’s quality of life, we advocate that SF should be the option of choice within the surgeon’s breast reconstruction techniques for selected cases.

## Figures and Tables

**Figure 1 jpm-11-01142-f001:**
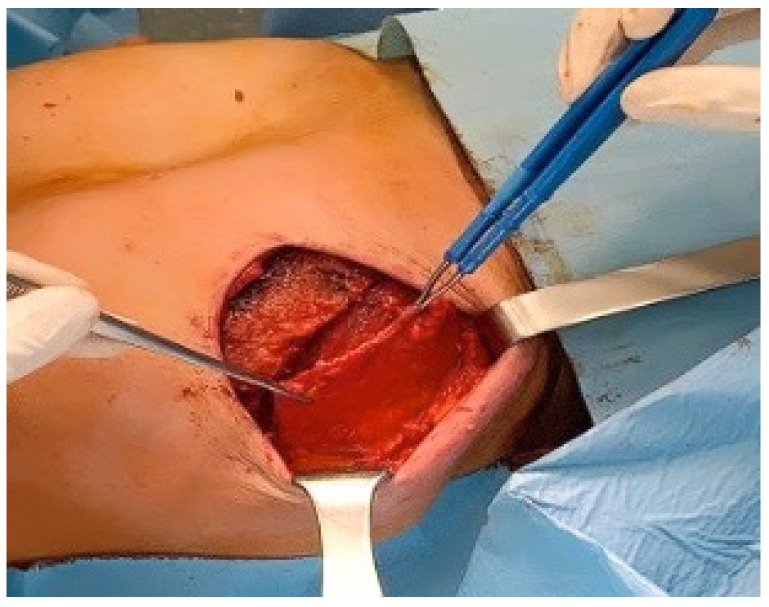
Serratus Fascia flap.

**Figure 2 jpm-11-01142-f002:**
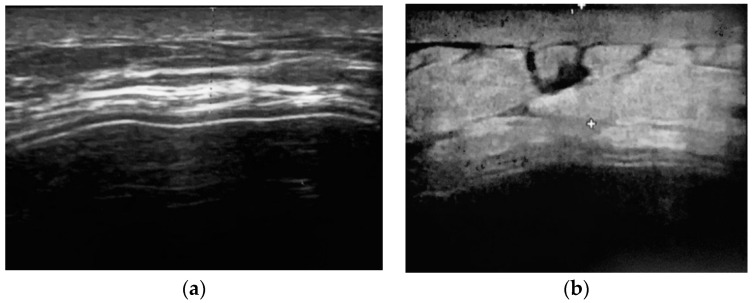
Example of US after breast reconstruction with a tissue expander using a muscolofascial pocket. (**a**) Thickness of the skin and subcutaneous tissue. (**b**) Total coverage thickness of the implant.

**Figure 3 jpm-11-01142-f003:**
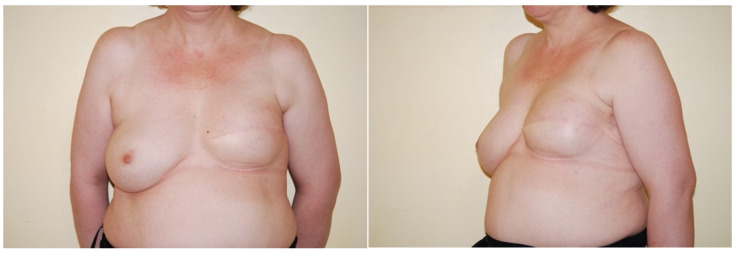
Six months post-op. Patient undergoing mastectomy and breast reconstruction with breast expander.

**Table 1 jpm-11-01142-t001:** Patient demographics. BMI: body max index, SSM: skin sparing mastectomy, NSM: Nipple-sparing mastectomy, CHT: chemotherapy, NEO: Neoadjuvant chemotherapy, RT: radiotherapy.

Patients Demographics								
No.	Age	BMI	Smoking	Mastectomy	CHT and RT	Drains Time	Follow-Up	Complications
1	57	28	-	SSM	/	18	15	-
2	41	26.5	-	SSM	/	12	15	-
3	38	23	-	SSM	CHT	11	15	-
4	44	23.51	-	SSM	NEO + RT	17	15	-
5	45	24	-	NSM	NEO	19	16	-
6	43	28	-	SSM	NEO	25	15	seroma
7	56	25	-	SSM	NEO	15	22	-
8	53	39	-	SSM	/	30	15	seroma
9	49	24	+	SSM	/	9	15	-
10	62	25	-	SSM	NEO	14	22	hematoma
11	59	18.75	+	SSM	NEO	11	12	-
12	59	19	+	SSM	NEO	15	12	-
13	42	39	-	SSM	NEO + RT	22	21	-
14	37	23	+	SSM	NEO	16	20	-
15	37	21	-	NSM	NEO	10	19	-
16	44	20	+	SSM	/	9	12	-
17	50	27.55	+	SSM	NEO	13	21	-
				SSM		11		-
18	52	17.48	+	SSM	NEO	15	23	-
				SSM		13		-

**Table 2 jpm-11-01142-t002:** US measurements, T-US: time from surgery to US examination, PPF: periprosthetic fluid, skin + sc: skin + subcutaneous thickness, SF: serratus fascia, TOT: Total thickness over the expander.

US Measurements					
No.	T-US (Months)	Skin + sc (mm)	SF + Capsule (mm)	TOT	US Evidences
1	4	6.8	7	13.8	PPF 20 × 11 mm
2	4	9	6	15	PPF 37 × 4 mm
3	4	8.5	6.5	15	PPF 11 × 3 mm
4	4	7.7	10.3	18	PPF 27 × 4 mm
5	5	11.6	4.4	16	PPF 17 × 3 mm
6	4	13	9	22	PPF 4 mm
7	11	10.5	5.5	16	-
8	4	10	3	13	PPF 1 mm
9	6	8.1	6.9	15	PPF 28 × 13 mm
10	14	6.6	10.4	17	PPF 19 × 0.5 mm
11	1	8.5	6.5	15	-
12	4	6.6	5.4	12	-
13	10	20	4	24	-
14	8	6.1	3.9	10	-
15	11	8.3	5.7	14	-
16	4	8.8	11.2	20	-
17	10	12.2	3.8	16	-
18	10	12.5	3.5	16	-
19	15	5	7	12	PPF 18 × 5 mm
20	15	4.5	7.5	12	PPF 30 × 17 mm

**Table 3 jpm-11-01142-t003:** VAS pain measurement: values of VAS the first and the third day after surgery.

Pain Intensity after Surgery on the VAS Scale		
No.	VAS I Day	VAS III Day
1	5	2
2	6	3
3	3	1
4	2	1
5	5	2
6	2	3
7	4	2
8	4	1
9	6	1
10	5	2
11	4	2
12	2	2
13	4	1
14	3	3
15	6	4
16	5	2
17	3	3
18	3	1
MEAN	4	2

## Data Availability

Not applicable.

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
