# Peer review of "Breast Reconstruction after Mastectomy with the Use of an Implant and Serratus Anterior Fascia Flap—Initial Clinical Evaluation"

_jpm, 2021, doi:10.3390/jpm11111142_

Round 1
Reviewer 1 Report
The paper presents a expander coverage in the lateral/caudal part of the breast. Please clarify
- Why did they use expander, and not direct to implant? Was so much skin resected?
- what kind of expander did they use.
- What were the co-morbidities of the patients (only mentioned, but not included in the list)
- Fig 1 does not show the inferior pectoral fascia. Was the expander/implant completely covered?
- please show long term follow up pictures with implant position, as this is the real outcome.
- Is this technique superior to direct to epipectoral /expander implant position?
- basically, the technique was described by Saint Cyr. what is new here?
Reviewer 2 Report
See annex
1 - I propose a title: Breast reconstruction after mastectomy with the use of an implant and serratus anterior fascia flap - initial clinical evaluation
18 - Write the abstract according to generally accepted rules: introduction, purpose, material and methods, conclusions (250 words). The content must match those later in the manuscript
33 - Keywords only partially reflect the topic of the work
57 - Not all items cited are content-related.
62 - ADM - describe abbreviation - Acellular dermal matrix
75 What stage of cancer advancement did the examined patients have?
79 - Describe the abbreviation: SS
109 - At the top, the table's title is without a description of the abbreviations used. Description of abbreviations below the table (in all tables}
114- What CHT had the 3rd patient
107 - (Table 1) a large discrepancy in the observations of 12 to 23 months may affect the evaluation of postoperative outcomes.
119 - What were the indications for the use of antibiotics?
155 - No table title
177- Better title is: Pain intensity after surgery on the VAS scale
235 - It is unnecessary to cite as many as 20 items of literature describing the same clinical problem
191 - The discussion is a literature review rather than presenting one's results against the results of other authors
260 - Conclusions do not follow directly from the results obtained. Instead, it is a summary of the surgical technique and early observation of patients in the material of our own and other authors.
272 - Too many references. Many repetitions and positions are not directly related to the topic of work.
The number of patients, the lack of randomization with another technique, and the relatively short period of observation limit the significance of the obtained results.

Author Response
We would like to thank the Reviewers for the meaningful suggestions on our paper entitled “Breast reconstruction after mastectomy with the use of an implant and serratus anterior fascia flap - initial clinical evaluation”. Their comments gave us the opportunity to improve our paper and we hope it will be suitable for publication, you will find all the revisions in the main text, evidenced in red.
- In the submission rules is written: The abstract should be a total of about 200 words maximum. The abstract should be a single paragraph and should follow the style of structured abstracts, but without headings.
- new keyword
68-ok
85 ok
109 – ok
114 cyclophosphamide, methotrexate 5-FU
119 ok
155 OK
177 OK
235 ok
272 ok

Round 2
Reviewer 1 Report
My questions were only answered in the cover letter. It is mandatory to cover them in the manuscript.
Author Response
We now covered the answer in the article.
Reviewer 2 Report
See annex

Author Response
We added some corrections in the manuscript.
Almost 85% were II stage breast cancer and 15% were III stage.
We added CHT protocol.
For the purpose of this study, follow up time wasn't considered a significant parameter, and the focus was on assessing the coverage of the implant before proceding to the second stage.
we added the limits of this study.